# ROBUST CORE-PERIPHERY CONSTRAINED TRANS-FORMER FOR DOMAIN ADAPTATION

## ABSTRACT

Unsupervised domain adaptation (UDA) aims to learn transferable representation across domains. Recently a few UDA works have successfully applied Transformer-based methods and achieved state-of-the-art (SOTA) results. However, it remains challenging when there exists a large domain gap between the source and target domain. Inspired by the remarkable transferability abilities of humans, where knowledge can adapt from familiar to uncharted domains, we endeavor to apply universally existing brain structure and function principles, specifically, the core-periphery principle and the concept of the noisy brain, to design and enhance the Transformer, ultimately improving its performance in UDA. In this work, we propose a novel brain-inspired robust core-periphery constrained transformer (RCCT) for unsupervised domain adaptation, which brings a large margin of performance improvement on various datasets. The application of the core-periphery principle and the development of the latent feature interaction (LFI) operation correspond to the 'Core-periphery' and 'Robust' aspects mentioned in the title. Specifically, in RCCT, the self-attention operation across image patches is rescheduled by an adaptively learned weighted graph with the Core-Periphery structure (CP graph), where the information communication and exchange between image patches are manipulated and controlled by the connection strength, i.e., edge weight of the learned weighted CP graph. In addition, considering the noisy nature of data in domain adaptation tasks, we propose a latent feature interaction operation to enhance model robustness, wherein we intentionally introduce interactions to the latent features in the latent space, ensuring the generation of robust learned weighted core-periphery graphs. We conducted extensive evaluations on several well-established UDA benchmarks, and the experimental results demonstrate that applying brain-inspired principles leads to promising results, surpassing the performance of existing Transformer-based methods.

## 1 INTRODUCTION

Deep neural networks (DNNs) made breakthroughs in various application fields due to their powerful automatic feature extraction capabilities. However, such impressive success usually needs great amounts of labeled data which can not be realized in the real case because of considerable time and expensive labor forces. Fortunately, unsupervised domain adaptation (UDA) Wilson & Cook (2020) techniques can leverage rich labeled data from the source domain and transfer knowledge from the source domain to the target domains with no or limited labeled examples. The key point of UDA is to find the discriminant and domain-invariant features from the labeled source domain and the unlabeled target domain in the common latent space. Along with more and more resources devoted to domain adaption research, the past decades have witnessed many UDA methods proposed and evolved Ganin & Lempitsky (2015) Liang et al. (2020) Long et al. (2018) Shu et al. (2018)Zhang et al. (2019).

However, the existing methods are all artificial neural network (ANN) driven structures, including variants of CNNs in conjunction with advanced techniques, such as adversarial learning, or the newest structures of Transformers combined with effective techniques like self-refinement Sun et al. (2022) Yang et al. (2023) Xu et al. (2022) Xu et al. (2019). More and more studies have found that the best-performing ANNs surprisingly resemble biological neural networks (BNN), which indicates that ANNs and BNNs may share common principles to achieve optimal performance in either

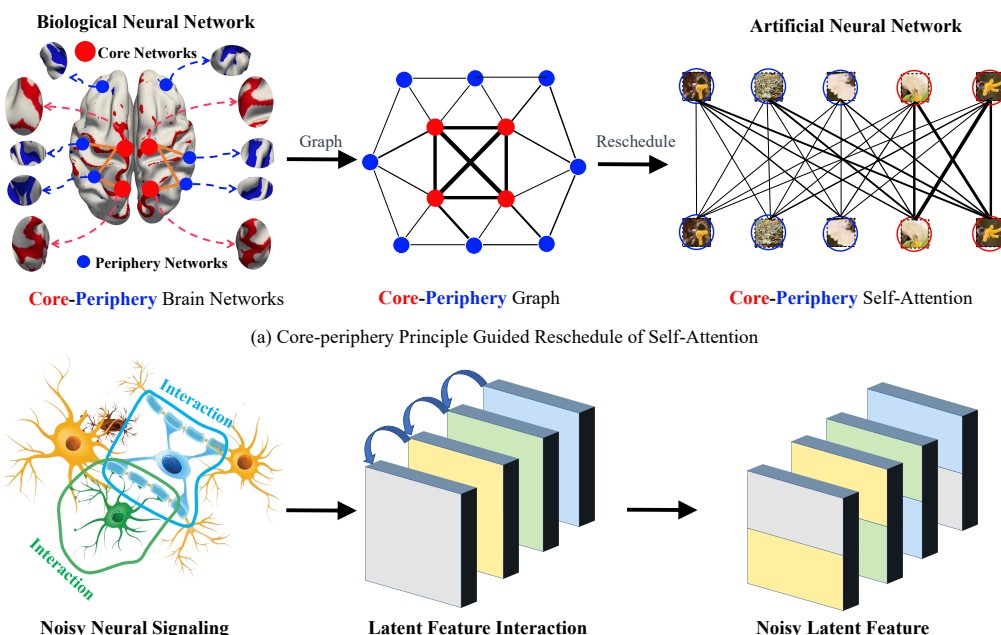

(a) Core-periphery Principle Guided Reschedule of Self-Attention

(b) Latent Feature Interaction Inspired by The Noisy Brain

Figure 1: Brain-inspired operations in model architecture and latent features. Part (a) shows the core-periphery principle guided redesign of self-attention in Transformers. The Core-Periphery structure broadly exists in brain networks, with a dense "core" of nodes (red) densely interconnected with each other and a sparse "periphery" of nodes (blue) sparsely connected to the core and among each other. Inspired by this principle of BNN, we aim to instill the Core-Periphery structure into the self-attention mechanism and propose a new core-periphery principle-constrained Transformer model for unsupervised domain adaptation. Part (b) illustrates the presence of noisy neural signaling in brain neurons. Drawing an analogy to this phenomenon within the noisy brain, we introduce a latent feature interaction operation that mimics the noisy signaling pattern.

machine learning or cognitive tasks You et al. (2020). Inspired by the studies in information communication, exchange, and processing in brain networks, in this work, we aim to proactively take advantage of the noisy neuro signaling to design a latent feature interaction operation and instill the organizational principle of Core-Periphery structure in BNNs to improve the domain adaptation ability of ANNs. The concepts of the Core-Periphery brain network and noisy brain are illustrated in Figure 1, where part (a) shows the connections between cores are much denser and stronger than the counterparts between peripheries, and part (b) shows the noisy nature of neuron signaling.

Aiming to bring brain-inspired priors into the ANNs, in this work, we propose a novel robust core-periphery constrained transformer (RCCT) for unsupervised domain adaptation. RCCT takes a vision transformer as the backbone and manipulates the strength of self-attention under the core-periphery constraints so that the information communication and exchange among the core patches are more effective and efficient while weakening the unimportant information flows among the periphery patches. Furthermore, by leveraging the concept of the noisy brain, we have designed a latent feature interaction operation to emulate this phenomenon. Specifically, RCCT has two key components that lead to its excellent performance, one is the core-periphery principle guided self-attention, and the other is the robust adaptive core-periphery graph generation realized by latent feature interaction.

We conclude the two key brain-inspired components:

• We have developed a novel Unsupervised Domain Adaptation (UDA) solution RCCT, which proactively incorporates the brain-inspired core-periphery principle to modulate the connection strength of self-attention in the vision transformer. This enhancement aims to bolster its ability to provide strong, transferable feature representations.

• We harness the concept of the noisy brain to design a Latent Feature Interaction (LFI) operation in tandem with the core-periphery constrained Transformer layer. This LFI operation functions in both the source and target domains, aiding in the creation of a robust core-periphery graph. The LFI empowers the model to adaptively learn domain-invariant core patches as well as domain-specific periphery patches.

## 2 RELATED WORKS

### 2.1 CORE-PERIPHERY STRUCTURE

The Core-Periphery structure is a fundamental network signature that is composed of two qualitatively distinct components: a dense "core" of nodes strongly interconnected with one another, allowing for integrative information processing to facilitate the rapid transmission of messages, and a sparse "periphery" of nodes sparsely connected to the core and among each other Gallagher et al. (2021). The Core-Periphery pattern has helped explain a broad range of phenomena in network-related domains, including online amplificationBarberá et al. (2015), cognitive learning processes Bassett et al. (2013), technological infrastructure organization Alvarez-Hamelin et al. (2005); Carmi et al. (2007), and critical disease-spreading conduits Kitsak et al. (2010). All these phenomena suggest that the Core-Periphery pattern may play a critical role in ensuring the effectiveness and efficiency of information exchange within the network.

### 2.2 THE NOISY BRAIN

Studies show that the human brain is noisy, generating electrical activity that includes random fluctuations, noise, and stored knowledge patterns Rolls & Deco (2010) Rolls & Deco (2013). This noise is often referred to as neural noise or neuronal variability, and it arises from the intrinsic properties of neurons and the complex interactions between them Hancock et al. (2017). Neural noise is a fundamental aspect of brain function and is not necessarily negative Faisal et al. (2008) Ferster (1996). In fact, it can play a role in various cognitive processes. For example, some studies suggest that neural noise may contribute to decision-making processes, creativity, and the brain's ability to explore different solutions to problems Haken (2006). Researchers in neuroscience and related fields continue to study neural noise to better understand its functional significance in brain function Averbeck et al. (2006). Encouraged by research on the noisy brain, we have designed a latent feature interaction (LFI) operation entangled with the core-periphery layer to fully transfer the knowledge learned from the human brain.

## 3 METHODS

### 3.1 PRELIMINARIES

In UDA, the images set in the labeled source domain are represented as $D_s \{(x_i^s, y_i^s)\}_{i=1}^{n_s}$, where $x_i^s$ are the images from the source domain, $y_i^s$ are the corresponding labels, and $n_s$ are the number of samples. The target domain is represented as $D_t \{(x_i^t)\}_{j=1}^{n_t}$ with $n_t$ samples and no labels. Unsupervised domain adaptation solutions aim to learn domain-invariant and domain-specific features to minimize domain discrepancies and achieve the desired prediction performance on unlabeled target data. In the introduction of our proposed RCCT, we will use 'core features' and 'periphery features' to describe domain-invariant and domain-specific characteristics, respectively.

The common practice is to design an objective function that jointly learns feature embeddings and a classifier. The objective function is formulated as

$$min\{L_{CE} \{x_s, y_s\} + \alpha L_{dis} \{x_s, x_t\}\} \quad (1)$$

where $L_{CE}$ is the standard cross-entropy loss supervised in the source domain, $L_{dis}$ is a transfer loss that incorporates various solutions, including the commonly used domain adversarial loss that promotes a domain-invariant feature space by employing a domain discriminator, and $\alpha$ is used to control the balance of $L_{dis}$.

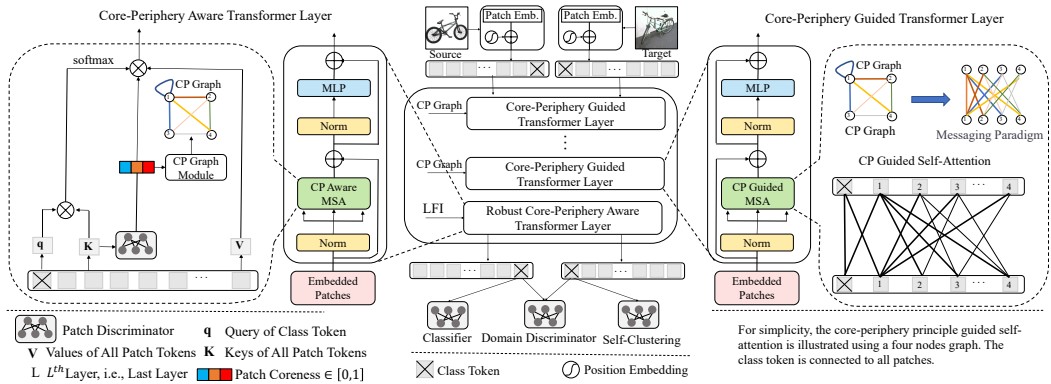

Figure 2: The overview of the proposed RCCT framework. In RCCT, source and target images are divided into non-overlapping fixed-size patches which are linearly projected into the latent space and concatenated with positional information. A class token is prepended to the image patches. The image patches and class token are delivered into a transformer encoder whose last layer is leveraged by a robust core-periphery aware layer, and the self-attention in previous layers is rescheduled by the adaptively core-periphery graph learned in the last layer. The core-periphery graphs are robustly learned through the process of latent feature interaction. Domain-invariant/Domain-specific feature learning is therefore contained in core patches/periphery patches. Adversarial domain adaptation is accomplished by patch-level and global-level discriminators.

## 3.2 METHODOLOGY

We aim to manipulate the strength of self-attention among dominant-invariant (core) and dominant-specific (periphery) patches by adaptively learning robust core-periphery graphs with latent feature interaction operation on both the source and target domains. Figure 2 illustrates the whole framework of the proposed RCCT. The RCCT network comprises a vision transformer backbone, a domain discriminator for the class token, a patch discriminator for patch tokens, a head classifier, a self-clustering module, and a core-periphery graph generation module, where the latent feature interaction operation is incorporated in the CP graph generation module. For images of each domain, the Patch Embedding layer linearly mapped them into a token sequence including a special class token and image tokens.

## 3.3 ROBUST CORE-PERIPHERY AWARE TRANSFORMER LAYER

As shown in Figure 2, we introduce the Robust Core-Periphery Aware Transformer Layer that takes advantage of the intrinsic merits of ViT, i.e., self-attention mechanisms and sequential patch tokens. Moreover, we leverage the self-attention mechanism with the CP principle and the noisy brain concept through the core-periphery aware module and latent feature interaction operation, separately.

### 3.3.1 CORE-PERIPHERY AWARE MODULE

The patch tokens correspond to partial regions of the image and capture visual features as fine-grained local representations. Existing work Yang et al. (2023) shows that the patch tokens are of different semantic importance, in this work, we define the coreness of the core-periphery principle to index the importance of patches, higher coreness patches are more likely to correspond to the domain-invariant patches, whereas lower coreness patches corresponding to domain-specific patches. Core-periphery aware layer aims at learning different coreness indices to those patch tokens for two purposes. First, to encourage the global image representation, i.e., the class token in the last layer, to attend to core tokens. Second, to strengthen the information communications among the core tokens and weaken the information among periphery tokens by rescheduling the self-attention under the guidance of the core-periphery graphs generated via the coreness of each patch token.

To obtain the coreness of patch tokens, we adopt a patch-level domain discriminator $D_l$ to evaluate the local features by optimizing:

$$L_{pat}\left(x^s, x^t\right) = -\frac{1}{nP} \sum_{x_i \in D} \sum_{p=1}^{P} L_{CE}\left(D_l\left(G_f\left(x_{ip}^*\right)\right), y_{ip}^d\right) \tag{2}$$

where $P$ is the number of patches, $D = D_s \cup D_t$, $G_f$ is the encoder for feature learning, implemented as ViT, $n = n_s + n_t$, is the total number of images of the source and target domain, the superscript $*$ denotes a patch from either source or target domain, $x_{ip}^*$ represents the $p$th of the $i$th image, $y_{ip}^d$ denotes the domain label of the $p$th token of the $i$th image, i.e., $y_{ip}^d = 1$ means source domain, else the target domain. $D\left(f_{ip}\right)$ gives the probability of the patch belonging to the source domain. During the training process, $D_l$ tries to discriminate the patches correctly, assigning 1 to patches from the source domain and 0 to those from the target domain, while $G_f$ combats such circumstances.

Empirically, patches that can easily deceive the patch dominator (e.g., $D_l$ is around 0.5) is more likely to be domain-invariant across domains and should be given a higher coreness. Therefore, we use $C\left(f_{ip}\right) = H\left(D_l\left(f_{ip}\right)\right) \in [0, 1]$ to measure the coreness of $r$th token of $i$th image, where $H\left(\cdot\right)$ is the standard entropy function. The explanation for the coreness is that by assigning an index to different patches, the model separates an image into domain-invariant representations and domain-specific representations, and the information communication from domain-specific features is softly suppressed. The generated core-periphery graph is then formulated as:

$$M_{cp} = \frac{1}{BH} \sum_{h=1}^{H} \sum_{b=1}^{B} \left[\left[C\left(f_{ip}\right)\right]^T C\left(f_{ip}\right)\right]_{\times} \tag{3}$$

$$M_{cp}\left(i, j\right) = \begin{cases} sqrt(M\left(i, j\right)) & \text{if } M\left(i, j\right) \geq 0.5 \\ square(M\left(i, j\right)) & \text{if } M\left(i, j\right) < 0.5 \end{cases} \tag{4}$$

where $T$ means transpose of the matrix, $B$ is the batch size, $H$ is the number of heads, $[\cdot]_{\times}$ means no gradients back-propagation for the adjacency matrix of the generated core-periphery graph, $sqrt(\cdot)$ and $square(\cdot)$ are the square root and square operations, respectively. The $sqrt(\cdot)$ and $square(\cdot)$ operations make the core-periphery property more apparent in the CP graph. The mask matrix $M_{cp}$ is the adjacency matrix of the core-periphery graph, and it defines the connection strength of the patch pairs. The connection strength among those patches with higher coreness is strengthened, and vice versa.

The vanilla MSA in the last layer can be redesigned by adopting the coreness of the patches, i.e., injecting the learned corners into the self-attention weights of the class token. As a result, the coreness aware self-attention (CSA) in the last transformer layer is defined as:

$$CSA(q, K, V) = softmax(\frac{qK^T}{\sqrt{d}}) \odot [1; C\left(K_{patch}\right)] V \tag{5}$$

where $q$ is the query of the class token, $K_{patch}$ is the key of the patch tokens, $\odot$ is the dot product, and $[; ]$ is the concatenation operation. Obviously, the CSA means that the class token takes more information from dominant-invariant patches with high coreness and hinders information from patches with low coreness. The coreness aware multi-head self-attention is therefore defined as:

$$C\text{-}MSA(q, K, V) = Concat(head_1, ..., head_k)W^O \tag{6}$$

where $head_i = CSA\left(qW_i^q, KW_i^K, VW_i^V\right)$. Taken them together, the operations in the last transformer layer are formulated as:

$$\begin{aligned} \hat{z}^l &= C\text{-}MSA\left(LN\left(z^{l-1}\right)\right) + z^{l-1} \\ z^l &= MLP\left(LN\left(\hat{z}^l\right)\right) + \hat{z}^l \end{aligned} \tag{7}$$

In this way, the core-periphery aware transformer layer focuses on fine-grained features that are dominant-invariant and are discriminative for classification. Here $l = L$, $L$ is the number of transformer layers in ViT architecture.

### 3.3.2 Latent Feature Interaction (LFI) on Dual-Domain

Evidence from neuroscience shows that the signal-processing process in the human brain is noisy, and influenced by potential noise or perturbations. More and more evidence also shows that noisy procedure in deep learning enhances model robustness Sun et al. (2022) Pereira et al. (2021). Drawing inspiration from the concept of the noisy brain in neuroscience that suggests that human brain signal processing is inherently noisy and susceptible to perturbations, and considering the growing body of evidence supporting the idea that adding noise to deep learning procedures enhances model robustness, we propose a Latent Feature Interaction (LFI) operation. This operation involves adding linearly transformed latent features to the core-periphery aware transformer layer. The LFI aims to enhance the stability and robustness of the generated CP graphs and make the model resistant to noisy perturbations. Actually, adding LFI to the core-periphery aware transformer layer imposes a regularization on multiple layers simultaneously.

Given an image $x_i$ either in the source domain or target domain, let $b_{x_i}$ be its input token sequence at the core-periphery aware transformer layer. $b_{x_i}$ is viewed as a representation of $x_i$ in the latent space. It is not effective to interact with latent features arbitrarily; instead, it is more beneficial to follow the pattern of noisy neuron signaling. Thus we use the token sequence $b_{x_j}$ of another sequentially chosen image $x_j$ from the same domain to add an offset. The LFI operation of $b_{x_i}$ can be formulated as:

$$\tilde{b}_{x_i} = (1 - \mu)b_{x_i} + \mu \left[ b_{x_j} \right]_\times, i \neq j \tag{8}$$

where $\mu$ is a scalar, controlling the feature interaction ratio, and $[\cdot]_\times$ means no gradient backpropagation. The LFI operation helps generate robust CP graphs.

### 3.4 Core-Periphery Guided Transformer Layer

With the representation paradigm, an unweighted complete graph can represent the self-attention of the vanilla ViT, and similarly, the core-periphery constraints can be effectively and conveniently infused into the ViT architecture by upgrading the complete graph with the generated weighted CP graphs, which is illustrated in the right part of Figure 2. Remember the generation process of the CP graph in the previous section, with the guidance of the CP graph, the first $L - 1$ transformer layer will focus on the likely core patches, i.e., dominant-invariant features, and suppress the information flow among periphery patches.

A CP graph can be represented by $\mathcal{G} = (\mathcal{V}, \mathcal{E})$, with nodes set $\mathcal{V} = \{\nu_1, ..., \nu_n\}$, edges set $\mathcal{E} \subseteq \{(\nu_i, \nu_j)|\nu_i, \nu_j \in \mathcal{V}\}$, and adjacency matrix $M_{cp}$. The CP graph guided self-attention for a specific patch $i$ at $r$-th layer of RCCT is defined as:

$$x_i^{(r+1)} = \sigma^{(r)}(\{(\frac{q_i^{(r)}(K_j^{(r)})^T}{\sqrt{d_k}})V_j^{(r)}, \forall j \in N(i)\}) \tag{9}$$

where $\sigma(\cdot)$ is the activation function, which is usually the softmax function in ViTs, $q_i^{(r)}$ is the query of patches in the $i$-th node in $\mathcal{G}$, $N(i) = \{i|i \vee (i, j) \in \mathcal{E}\}$ are the neighborhood nodes of node $i$, $d_k$ is the dimension of queries and keys, and $K_j^{(r)}$ and $V_j^{(r)}$ are the key and value of patches in node $j$. Therefore, the CP graph guided self-attention that is conducted at the patch level can be formulated as:

$$Attention(Q, K, V, M_{cp}) = softmax(\frac{QK^T \odot M_{cp}}{\sqrt{d_k}}V) \tag{10}$$

where queries, keys, and values of all patches are packed into matrices $Q$, $K$, and $V$, respectively, $M_{cp}$ is the adjacency matrix provided by the last transformer layer. Similar to the multi-head attention in transformers, our proposed CP-guided multi-head attention is formulated as:

$$MSA(Q, K, V, M_{cp}) = Concat(head_1, ..., head_h)W^o \tag{11}$$

where $head_i = Attention(QW_i^Q, KW_i^K, VW_i^V, M_{cp})$ where the parameter matrices $W_i^Q$, $W_i^K$, $W_i^V$ and $W^O$ are the projections. Multi-head attention helps the model to jointly aggregate information from different representation subspaces at various positions. In this work, we apply the CP constraints to each representation subspace.

Table 1: Comparison with SOTA methods on **Office-Home**. The best performance is marked in red.

| Method | Ar→Cl | Ar→Pr | Ar→Re | Cl→Ar | Cl→Pr | Cl→Re | Pr→Ar | Pr→Cl | Pr→Re | Re→Ar | Re→Cl | Re→Pr | Avg. |
|---|---|---|---|---|---|---|---|---|---|---|---|---|---|
| ResNet-50 | 44.9 | 66.3 | 74.3 | 51.8 | 61.9 | 63.6 | 52.4 | 39.1 | 71.2 | 63.8 | 45.9 | 77.2 | 59.4 |
| MinEnt | 51.0 | 71.9 | 77.1 | 61.2 | 69.1 | 70.1 | 59.3 | 48.7 | 77.0 | 70.4 | 53.0 | 81.0 | 65.8 |
| SAFN | 52.0 | 71.7 | 76.3 | 64.2 | 69.9 | 71.9 | 63.7 | 51.4 | 77.1 | 70.9 | 57.1 | 81.5 | 67.3 |
| CDAN+E | 54.6 | 74.1 | 78.1 | 63.0 | 72.2 | 74.1 | 61.6 | 52.3 | 79.1 | 72.3 | 57.3 | 82.8 | 68.5 |
| DCAN | 54.5 | 75.7 | 81.2 | 67.4 | 74.0 | 76.3 | 67.4 | 52.7 | 80.6 | 74.1 | 59.1 | 83.5 | 70.5 |
| BNM | 56.7 | 77.5 | 81.0 | 67.3 | 76.3 | 77.1 | 65.3 | 55.1 | 82.0 | 73.6 | 57.0 | 84.3 | 71.1 |
| SHOT | 57.1 | 78.1 | 81.5 | 68.0 | 78.2 | 78.1 | 67.4 | 54.9 | 82.2 | 73.3 | 58.8 | 84.3 | 71.8 |
| ATDOC-NA | 58.3 | 78.8 | 82.3 | 69.4 | 78.2 | 78.2 | 67.1 | 56.0 | 82.7 | 72.0 | 58.2 | 85.5 | 72.2 |
| ViT-B | 54.7 | 83.0 | 87.2 | 77.3 | 83.4 | 85.6 | 74.4 | 50.9 | 87.2 | 79.6 | 54.8 | 88.8 | 75.5 |
| TVT-B | 74.9 | 86.8 | 89.5 | 82.8 | 88.0 | 88.3 | 79.8 | 71.9 | 90.1 | 85.5 | 74.6 | 90.6 | 83.6 |
| CDTrans-B | 68.8 | 85.0 | 86.9 | 81.5 | 87.1 | 87.3 | 79.6 | 63.3 | 88.2 | 82.0 | 66.0 | 90.6 | 80.5 |
| SSRT-B | 75.2 | 89.0 | 91.1 | 85.1 | 88.3 | 90.0 | 85.0 | 74.2 | 91.3 | 85.7 | 78.6 | 91.8 | 85.4 |
| CCT-B | 77.6 | 89.6 | 90.7 | 85.0 | 89.3 | 89.7 | 84.4 | 74.6 | 91.9 | 86.6 | 77.0 | 91.8 | 85.7 |
| **RCCT-B** | **80.1** | **91.4** | **92.9** | **87.9** | **92.2** | **92.2** | **86.3** | **79.5** | **93.1** | **88.9** | **81.0** | **93.8** | **88.3** |

## 3.5 OVERALL OBJECTIVE FUNCTION

Since our proposed RCCT has a classifier, a self-clustering module, a patch discriminator, and a global discriminator, there are four terms in the overall objective function. The classification loss term is formulated as:

$$L_{clc}\left(x^s, y^s\right) = \frac{1}{n_s} \sum_{x_i \in D_s} L_{CE}\left(G_c\left(G_f\left(x_i^s\right)\right), y_i^s\right) \tag{12}$$

where $G_c$ is the classifier.

The domain discriminator takes the class token and tries to discriminate the class token, i.e., the representation of the entire image, to the source or target domain. The domain adversarial loss term is formulated as:

$$L_{dis}\left(x^s, x^t\right) = -\frac{1}{n} \sum_{x_i \in D} L_{ce}\left(D_g\left(G_f\left(x_i^*\right), y_i^d\right)\right) \tag{13}$$

where $Dg$ is the domain discriminator, and $y_i^d$ is the the domain label ((i.e., $y_i^d = 1$ means source domain, $y_i^d = 0$ is target).

The self-clustering module (SCM) is inspired by the cluster assumption Chapelle & Zien (2005) and the probability $p^t = softmax\left(G_c\left(G_f\left(x^t\right)\right)\right)$ of target image $x_t$ is optimized to maximize the mutual information with $x_t$ Yang et al. (2023). The self-clustering loss term is formulated as:

$$I\left(p^t; x^t\right) = H\left(\bar{p^t}\right) - \frac{1}{n_t} \sum_{i=1}^{n_t} H\left(p_i^t\right) \tag{14}$$

where $p_i^t = softmax\left(G_c\left(G_f\left(x_i^t\right)\right)\right)$ and $\bar{p^t} = \mathbb{E}\left[p^t\right]$. The self-clustering loss encourages the model to learn clustered target features

Take classification loss (Eq. 12), domain adversarial loss (Eq. 13), patch adversarial loss (Eq. 2), and self-clustering loss (Eq. 14) together, the overall objective function is therefore formulated as:

$$L_{clc}\left(x^s, y^s\right) + \alpha L_{dis}\left(x^s, x^t\right) + \beta L_{pat}\left(x^s, x^t\right) - \gamma I\left(p^t; x^t\right) \tag{15}$$

where $\alpha$, $\beta$, and $\gamma$ are the hyperparameters that control the influence of subterms on the overall function.

## 4 EXPERIMENTS

We evaluate our proposed RCCT on the most widely used UDA benchmarks, including **Office-31** Saenko et al. (2010), **Office-Home** Venkateswara et al. (2017), **VisDA2017** Peng et al. (2017),

Table 2: Comparison with SOTA methods on **Visda2017**. The best performance is marked in red.

| Method | plane | bcycl | bus | car | horse | knife | mcycl | person | plant | sktbrd | train | truck | Avg. |
|---|---|---|---|---|---|---|---|---|---|---|---|---|---|
| ResNet-50 | 55.1 | 53.3 | 61.9 | 59.1 | 80.6 | 17.9 | 79.7 | 31.2 | 81.0 | 26.5 | 73.5 | 8.5 | 52.4 |
| DANN | 81.9 | 77.7 | 82.8 | 44.3 | 81.2 | 29.5 | 65.1 | 28.6 | 51.9 | 54.6 | 82.8 | 7.8 | 57.4 |
| MinEnt | 80.3 | 75.5 | 75.8 | 48.3 | 77.9 | 27.3 | 69.7 | 40.2 | 46.5 | 46.6 | 79.3 | 16.0 | 57.0 |
| SAFN | 93.6 | 61.3 | 84.1 | 70.6 | 94.1 | 79.0 | 91.8 | 79.6 | 89.9 | 55.6 | 89.0 | 24.4 | 76.1 |
| CDAN+E | 85.2 | 66.9 | 83.0 | 50.8 | 84.2 | 74.9 | 88.1 | 74.5 | 83.4 | 76.0 | 81.9 | 38.0 | 73.9 |
| BNM | 89.6 | 61.5 | 76.9 | 55.0 | 89.3 | 69.1 | 81.3 | 65.5 | 90.0 | 47.3 | 89.1 | 30.1 | 70.4 |
| CGDM | 93.7 | 82.7 | 73.2 | 68.4 | 92.9 | 94.5 | 88.7 | 82.1 | 93.4 | 82.5 | 86.8 | 49.2 | 82.3 |
| SHOT | 94.3 | 88.5 | 80.1 | 57.3 | 93.1 | 93.1 | 80.7 | 80.3 | 91.5 | 89.1 | 86.3 | 58.2 | 82.9 |
| ViT-B | 97.7 | 48.1 | 86.6 | 61.6 | 78.1 | 63.4 | 94.7 | 10.3 | 87.7 | 47.7 | 94.4 | 35.5 | 67.1 |
| TVT-B | 92.9 | 85.6 | 77.5 | 60.5 | 93.6 | 98.2 | 89.4 | 76.4 | 93.6 | 92.0 | 91.7 | 55.7 | 83.9 |
| CDTrans-B | 97.1 | 90.5 | 82.4 | 77.5 | 96.6 | 96.1 | 93.6 | **88.6** | **97.9** | 86.9 | 90.3 | **62.8** | 88.4 |
| SSRT-B | **98.9** | 87.6 | **89.1** | **84.8** | 98.3 | **98.7** | **96.3** | 81.1 | 94.9 | **97.9** | 94.5 | 43.1 | 88.8 |
| CCT-B | 97.1 | 92.9 | 78.0 | 64.1 | 97.5 | 96.5 | 90.6 | 78.0 | 91.2 | 95.6 | 93.8 | 65.6 | 86.7 |
| RCCT-B | 98.4 | **95.9** | 87.7 | 77.3 | **98.9** | 96.7 | 95.8 | 82.6 | 96.4 | **97.9** | **97.8** | 62.8 | **90.7** |

and **DomainNet** Peng et al. (2019). We use the ViT-base with a $16 \times 16$ patch size (ViT-B/16) Dosovitskiy et al. (2020) Steiner et al. (2021), pre-trained on ImageNet-21k Russakovsky et al. (2015), as our vision transformer backbone. Details of datasets and settings are provided in the supplementary.

Table 3: Comparison with SOTA methods on **DomainNet**. The best performance is marked in red.

| ResNet101 | clp | inf | pnt | qdr | rel | skt | Avg. | MIM | clp | inf | pnt | qdr | rel | skt | Avg. | CGDM | clp | inf | pnt | qdr | rel | skt | Avg. |
|---|---|---|---|---|---|---|---|---|---|---|---|---|---|---|---|---|---|---|---|---|---|---|---|
| clp | - | 19.3 | 37.5 | 11.1 | 52.2 | 41.1 | 32.2 | clp | - | 15.1 | 35.6 | 10.7 | 51.5 | 43.1 | 31.2 | clp | - | 16.9 | 35.3 | 10.8 | 53.5 | 36.9 | 30.7 |
| inf | 30.2 | - | 31.2 | 3.6 | 44.0 | 27.9 | 27.4 | inf | 32.1 | - | 31.0 | 2.9 | 48.5 | 31.0 | 29.1 | inf | 27.8 | - | 28.2 | 4.4 | 48.2 | 22.5 | 26.2 |
| pnt | 39.6 | 18.7 | - | 4.9 | 54.5 | 36.3 | 30.8 | pnt | 40.1 | 14.7 | - | 4.2 | 55.4 | 36.8 | 30.2 | pnt | 37.7 | 14.5 | - | 4.6 | 59.4 | 33.5 | 30.0 |
| qdr | 7.0 | 0.9 | 1.4 | - | 4.1 | 8.3 | 4.3 | qdr | 18.8 | 3.1 | 5.0 | - | 16.0 | 13.8 | 11.3 | qdr | 14.9 | 1.5 | 6.2 | - | 10.9 | 10.2 | 8.7 |
| rel | 48.4 | 22.2 | 49.4 | 6.4 | - | 38.8 | 33.0 | rel | 48.5 | 19.0 | 47.6 | 5.8 | - | 39.4 | 22.1 | rel | 49.4 | 20.8 | 47.2 | 4.8 | - | 38.2 | 32.0 |
| skt | 46.9 | 15.4 | 37.0 | 10.9 | 47.0 | - | 31.4 | skt | 51.7 | 16.5 | 40.3 | 12.3 | 53.5 | - | 34.9 | skt | 50.1 | 16.5 | 43.7 | 11.1 | 55.6 | - | 35.4 |
| Avg. | 34.4 | 15.3 | 31.3 | 7.4 | 40.4 | 30.5 | 26.6 | Avg. | 38.2 | 13.7 | 31.9 | 7.2 | 45.0 | 32.8 | 28.1 | Avg. | 36.0 | 14.0 | 32.1 | 7.1 | 45.5 | 28.3 | 27.2 |
| MDD | clp | inf | pnt | qdr | rel | skt | Avg. | ViT | clp | inf | pnt | qdr | rel | skt | Avg. | CDTrans | clp | inf | pnt | qdr | rel | skt | Avg. |
| clp | - | 20.4 | 43.3 | 15.2 | 59.3 | 46.5 | 36.9 | clp | - | 27.2 | 53.1 | 13.2 | 71.2 | 53.3 | 43.6 | clp | - | 29.4 | 57.2 | 26.0 | 72.6 | 58.1 | 48.7 |
| inf | 32.7 | - | 34.5 | 6.3 | 47.6 | 29.2 | 30.1 | inf | 51.4 | - | 49.3 | 4.0 | 66.3 | 41.1 | 42.4 | inf | 57.0 | - | 54.4 | 12.8 | 69.5 | 48.4 | 48.4 |
| pnt | 46.4 | 19.9 | - | 8.1 | 58.8 | 42.9 | 35.2 | pnt | 53.1 | 25.6 | - | 4.8 | 70.0 | 41.8 | 39.1 | pnt | 62.9 | 27.4 | - | 15.8 | 72.1 | 53.9 | 46.4 |
| qdr | 31.1 | 6.6 | 18.0 | - | 28.8 | 22.0 | 21.3 | qdr | 30.5 | 4.5 | 16.0 | - | 27.0 | 19.3 | 19.5 | qdr | 44.6 | 8.9 | 29.0 | - | 42.6 | 28.5 | 30.7 |
| rel | 55.5 | 23.7 | 52.9 | 9.5 | - | 45.2 | 37.4 | rel | 58.4 | 29.0 | 60.0 | 6.0 | - | 45.8 | 39.9 | rel | 66.2 | 31.0 | 61.5 | 16.2 | - | 52.9 | 45.6 |
| skt | 55.8 | 20.1 | 46.5 | 15.0 | 56.7 | - | 38.8 | skt | 63.9 | 23.8 | 52.3 | 14.4 | 67.4 | - | 44.4 | skt | 69.0 | 29.6 | 59.0 | 27.2 | 72.5 | - | 51.5 |
| Avg. | 44.3 | 18.1 | 39.0 | 10.8 | 50.2 | 37.2 | 33.3 | Avg. | 51.5 | 22.0 | 46.1 | 8.5 | 60.4 | 40.3 | 38.1 | Avg. | 59.9 | 25.3 | 52.2 | 19.6 | 65.9 | 48.4 | 45.2 |
| SSRT | clp | inf | pnt | qdr | rel | skt | Avg. | CCT | clp | inf | pnt | qdr | rel | skt | Avg. | RCCT | clp | inf | pnt | qdr | rel | skt | Avg. |
| clp | - | 33.8 | 60.2 | 19.4 | 75.8 | 59.8 | 49.8 | clp | - | 30.6 | 56.9 | 17.8 | 69.8 | 58.0 | 46.6 | clp | - | 32.4 | 60.2 | 21.1 | 78.5 | 63.2 | 51.1 |
| inf | 55.5 | - | 54.0 | 9.0 | 68.2 | 44.7 | 46.3 | inf | 53.9 | - | 47.6 | 9.3 | 69.2 | 45.0 | 45.0 | inf | 57.5 | - | 55.8 | 9.7 | 71.6 | 47.8 | 48.5 |
| pnt | 61.7 | 28.5 | - | 8.4 | 71.4 | 55.2 | 45.0 | pnt | 52.5 | 26.2 | - | 8.4 | 70.0 | 48.0 | 41.0 | pnt | 63.5 | 29.4 | - | 9.4 | 72.5 | 54.9 | 45.9 |
| qdr | 42.5 | 8.8 | 24.2 | - | 37.6 | 33.6 | 29.3 | qdr | 37.6 | 10.5 | 19.6 | - | 29.3 | 26.9 | 24.8 | qdr | 42.2 | 12.4 | 23.6 | - | 33.8 | 30.6 | 28.5 |
| rel | 69.9 | 37.1 | 66.0 | 10.1 | - | 58.9 | 48.4 | rel | 63.9 | 32.4 | 61.7 | 11.6 | - | 53.4 | 44.6 | rel | 70.4 | 34.3 | 67.3 | 12.9 | - | 57.8 | 48.5 |
| skt | 70.6 | 32.8 | 62.2 | 21.7 | 73.2 | - | 52.1 | skt | 67.3 | 28.9 | 60.0 | 20.5 | 71.5 | - | 49.6 | skt | 72.6 | 31.9 | 64.1 | 22.1 | 75.4 | - | 53.2 |
| Avg. | 60.0 | 28.2 | 53.3 | 13.7 | 65.3 | 50.4 | 45.2 | Avg. | 55.0 | 25.7 | 49.2 | 13.5 | 62.0 | 46.3 | 42.0 | Avg. | 61.2 | 28.1 | 54.2 | 15.0 | 66.4 | 50.9 | **46.0** |

## 4.1 RESULTS

Table 1 presents evaluation results on the dataset Office-Home. The "-B" indicates results using ViT-base backbones. RCCT means robust core-periphery constrained transformer, whereas CCT means omitting LFI operation. The methods above the black line are based on CNN architecture, while those under the black line are developed from the Transformer architecture. Table 2 shows results on the dataset VisDA2017. The experimental results on the large dataset DomainNet are shown in Table 3. The core-periphery principle enables the model to outperform the ViT baseline, while the LFI operation enhances the model's performance compared to the most advanced methods. The RCCT also achieved the SOTA results on Office-31, as shown in Table 4. From the comparisons, the transformer-based methods gain much better results than CNN-based models thanks to their strong transferable feature representations. Compared with other methods, our RCCT-B performs the best

Table 4: Comparison with SOTA methods on **Office-31**. The best performance is marked in red.

| Method | A→W | D→W | W→D | A→D | D→A | W→A | Avg. |
|--------|-----|-----|-----|-----|-----|-----|------|
| ResNet-50 | 68.4 | 96.7 | 99.3 | 68.9 | 62.5 | 60.7 | 76.1 |
| DANN | 82.0 | 96.9 | 99.1 | 79.7 | 68.2 | 67.4 | 82.2 |
| rRGrad+CAT | 94.4 | 98.0 | 100.0 | 90.8 | 72.2 | 70.2 | 87.6 |
| SAFN+ENT | 90.1 | 98.6 | 99.8 | 90.7 | 73.0 | 70.2 | 87.1 |
| CDAN+TN | 95.7 | 98.7 | 100.0 | 94.0 | 73.4 | 74.2 | 89.3 |
| TAT | 92.5 | 99.3 | 100.0 | 93.2 | 73.1 | 72.1 | 88.4 |
| SHOT | 90.1 | 98.4 | 99.9 | 94.0 | 74.7 | 74.3 | 88.6 |
| MDD+SCDA | 95.3 | 99.0 | 100.0 | 95.4 | 77.2 | 75.9 | 90.5 |
| ViT-B | 91.2 | 99.2 | 100.0 | 93.6 | 80.7 | 80.7 | 91.1 |
| TVT-B | 96.4 | 99.4 | 100.0 | 96.4 | 84.9 | 86.1 | 93.9 |
| CDTrans-B | 96.7 | 99.0 | 100.0 | 97.0 | 81.1 | 81.9 | 92.6 |
| SSRT-B | **97.7** | 99.2 | 100.0 | **98.6** | 83.5 | 82.2 | 93.5 |
| CCT | 96.0 | **99.5** | 100.0 | 94.4 | 84.5 | 85.1 | 93.3 |
| RCCT | 97.4 | **99.5** | 100.0 | 96.4 | **88.1** | **88.7** | **95.0** |

on Office-Home, Office-31, DomainNet, and VisDA2017. Even the weaker version of CCT-B can surpass most methods on these datasets.

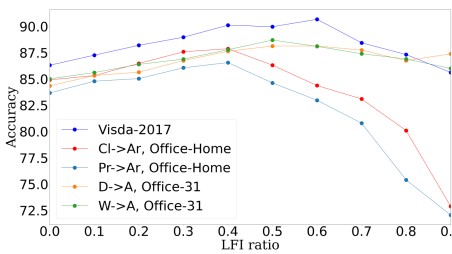

Figure 3: The influence of LFI ratio $\mu$ on accuracy. The ratio ranges from [0.0, 0.9].

Table 5: Ablation study of each module. The baseline is the ViT-base, SCM is the self-clustering module, LFI is the latent feature interaction operation, and CP is the core-periphery constraint.

| Methods | Office-31 | Office-Home | VisDA-2017 | DomainNet | Avg. |
|---------|-----------|-------------|------------|-----------|------|
| Baseline | 91.1 | 75.5 | 67.1 | 38.1 | 68.0 |
| +SCM | 93.0 | 84.9 | 85.5 | 39.6 | 75.8 |
| +CP | 93.3 | 85.7 | 86.7 | 42.0 | 76.9 |
| +LFI | 95.0 | 88.3 | 90.7 | 46.0 | 80.0 |

## 4.2 ABLATION STUDIES

We conduct ablation studies on the LFI ratio $\mu$. Figure 3 plots the influence of the LFI ratio $\mu$ on classification accuracy. Note when the coefficient ratio is 0, which means there is no LFI operation, the RCCT is degraded to CCT. From Figure 3, we can observe that RCCT can gain prediction improvements on a wide range of LFI ratios. We further evaluate the influence of different ingredients, including CP constraints, LFI operations, and SCM modules, on the domain adaptation tasks. The results are shown in Table 5. As shown in the table, all the ingredients contribute to the state-of-the-art (SOTA) results achieved by the proposed RCCT. Based on the results in Table 5, we observe that the brain-inspired principles, including the concept of the noisy brain (LFI) and core-periphery organization (CP), lead to significant performance improvements of the Transformer compared to the existing baseline.

## 5 CONCLUSION

In this paper, we propose a novel brain-inspired approach, named RCCT for unsupervised domain adaptation. It practically instills the core-periphery constraint into the self-attention in the Transformer architecture, along with a latent feature interaction (LFI) operation inspired by the concept of the noisy brain. The RCCT can adaptively learn core-periphery graphs by measuring the coreness of patches via a patch discriminator. At the same time, deliberate LFI operations are added to force the model to learn robust CP graphs. We use the learned CP graphs to manipulate self-attention weights to strengthen the information communication among higher coreness patches while suppressing that among low coreness patches. Our RCCT achieves promising results on four popular UDA datasets, demonstrating that brain-inspired ANNs hold significant potential in the field of UDA.

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
