# Supplementary for Robust Core-Periphery Constrained Transformer for Domain Adaptation

## 1 Datasets

We evaluate our proposed RCCT on the most widely used UDA benchmarks, including **Office-31** Saenko et al. (2010), **Office-Home** Venkateswara et al. (2017), **VisDA2017** Peng et al. (2017), and **DomainNet** Peng et al. (2019). **Office-31** Saenko et al. (2010) contains 4,652 images of 31 categories collected from three domains, i.e., Amazon (A), DSLR (D), and Webcam (W). **Office-Home** Venkateswara et al. (2017) has 15,500 images of 65 classes from four domains: Artistic (Ar), Clip Art (Cl), Product (Pr), and Real-world (Rw) images. **VisDA2017** Peng et al. (2017) is a Synthetic-to-Real object recognition dataset, with more than 0.2 million images in 12 classes. **DomainNet** Peng et al. (2019) dataset has the largest scale containing around 0.6 million images of 345 classes in 6 domains: Quickdraw (qdr), Real (rel), Sketch (skt), Clipart (clp), Infograph (inf), Painting (pnt).

## 2 Training Details

The ViT-B/16 contains 12 transformer layers in total. We use minibatch Stochastic Gradient Descent (SGD) optimizer Ruder (2016) with a momentum of 0.9 as the optimizer. The batch size is set to 32 for all the experiments. We initialized the learning rate as 0 and linearly warm up to 0.06 after 500 training steps. We then schedule it using the cosine decay strategy. For small to middle-scale datasets such as Office-31 and Office-Home, the epoch is set to 5000. For large-scale datasets Visda-2017 and DomainNet, the epoch is set to 20000. The LFI coefficient $\mu$ is set as 0.5, but we have conducted an ablation study that explores values within the range of $(0, 1.0)$. The hyper-parameters $\alpha$, $\beta$, and $\gamma$ are set to $[1.0, 0.01, 0.1]$ for Office-31 and Office-Home, $[0.1, 0.1, 0.1]$ for Visda-2017 and DomainNet.

## 3 Details of Core-Periphery Implementation

The patch discriminator will evaluate the importance of image patches. A patch that can easily fool the patch discriminator, indicating it is more likely domain-invariant, is considered to have high importance, and vice versa. We use the term 'coreness' to define the importance of image patches. Therefore, by knowing the coreness of image patches and considering each image patch as a node, we can generate a weighted graph where the edge weights are determined by the coreness of the patches. For example, for simplicity, given two image patches, $p1$ and $p2$, the patch discriminator assigns coreness values of 0.9 and 0.5 to the two image patches, respectively. The weight of the edge between $p1$ and $p2$ is calculated as $0.9 \times 0.5$, the weight of the self-loop edge for p1 is calculated as $0.9 \times 0.9$, and the weight of the self-loop edge for p2 is calculated as $0.5 \times 0.5$.

For the initial epoch, since we do not know the coreness of each image patch. Therefore, we initiate with an unweight complete graph to guide self-attention. With each iteration, the patch discriminator in the robust core-periphery aware transformer layer will asses the coreness of each image patch. The patch discriminator also encourages the class token in the last transformer layer to focus on dominant-invariant features in core patches and contempt the dominant-specific features in periphery nodes. Then the CP graph module will generate a CP graph according to the patch coreness. In the meantime, we have incorporated a latent feature interaction operation into the core-periphery aware transformer layer. This addition not only helps manifest the concept of the 'noisy brain' but also

fortifies the model against potential fluctuations. From the second epoch, the transformer layers before the robust CP aware layer will adopt the generated CP graph to reschedule the self-attention to strengthen the information communication among core patches (dominant-invariant) and weaken the connection among periphery patches (dominant-specific).

The classifier takes the class token of the source domain images and outputs label prediction. The domain discriminator takes the output class tokens of the source and target domain to be aligned in the latent space by playing a two-player min-max game with the feature extractor. The self-clustering module enforces the aligned features of different classes of target-domain images to be clustered and separable.

## 3.1 LEARNED CORE-PERIPHERY GRAPHS

Our method adaptively learns core-periphery graphs for different datasets and tasks and uses the learned core-periphery graphs to reschedule self-attention. Some randomly selected learned core-periphery graphs in the format of adjacency matrices are shown in Figure 1. The first line includes the learned CP graphs from CCT, while the second line shows the CP graphs learned from RCCT. These graphs have been examined by core-periphery detection algorithmsKojaku & Masuda (2017). Obviously, the CP graphs from RCCT show more dense and weighted patterns, which helps capture the core patches from different domains and improves domain adaptivity across domains. The visualization of learned CP graphs demonstrates the effectiveness of the core-periphery principle and the concept of noisy brain in domain adaptation tasks.

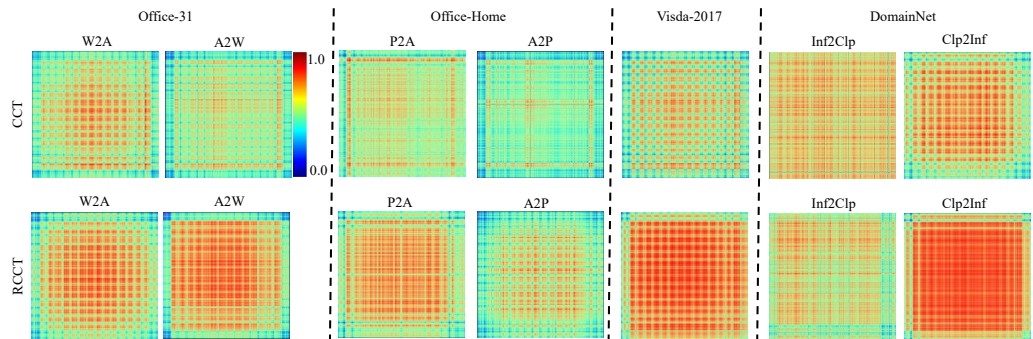

Figure 1: The learned core-periphery graphs (adjacency matrics) from some randomly selected domain adaptation tasks. The first line includes the CP graphs generated from the CCT model, while the second lines are the CP graphs from RCCT mode. The texts above the CP graphs show the task to which the CP graphs belong. The redder colors the higher the weight.