# OpenReview forum: "Robust Core-periphery Constrained Transformer for Domain Adaptation"
_ICLR.cc/2024/Conference — ICLR 2024 Conference Withdrawn Submission_

### Official Review · Reviewer_dFo7 · 2023-10-24

**Soundness:** 3 good
**Presentation:** 3 good
**Contribution:** 3 good
**Rating:** 6
**Confidence:** 3

**Summary:**

This paper targets on unsupervised domain adaptation. It applies the core-periphery principle to enhance the Transformer, improving its performance on unsupervised domain adaptation. Concretely, the proposed robust core-periphery constrained transformer (RCCT) brings about large performance improvements on various datasets.

**Strengths:**

1. This paper is well-written and easy to follow.
2. The experiment results is strong, proving the effectiveness of the proposed method.

**Weaknesses:**

1. Why the improvement appears to be smaller on DomainNet than other datasets? On DomainNet, RCCT: 46, CDTrans: 45.2, while on Visda, RCCT: 90.7, CDTrans: 88.4. I know DomainNet is more difficult to handle, but we still can have some further discussions.
2. I am also curious about the performance on other domain adaptation settings, such as Partial Domain Adaptation. Can we have a discussion?

**Questions:**

See weakness.

---

### Official Review · Reviewer_4j1x · 2023-10-30

**Soundness:** 1 poor
**Presentation:** 2 fair
**Contribution:** 1 poor
**Rating:** 1
**Confidence:** 4

**Summary:**

This paper proposes a  brain-inspired robust core-periphery constrained transformer (RCCT) for unsupervised domain adaptation. In RCCT, self-attention across image patches is realized by an adaptively learned weighted graph with the Core-Periphery structure. In addition, a latent feature interaction operation is introduced to enhance the fusion of tokens. Some empirical results are provided to validate the effectiveness of RCCT.

**Strengths:**

1. The empirical performance of RCCT is strong.

**Weaknesses:**

1. The novelty of this paper may be insufficient. The "core-periphery" attention is simply a slightly modified standard multi-head self-attention. "Latent feature interaction" seems to be a simple feature-space mix-up. None of them are novel. Their connection to the biological brain is simply far-fetched.

2. All the citation formats in the paper are incorrect. Many equations include errors, e.g., Eqs. (3-4).

3. As shown in Eq. (15), the proposed method introduces too many additional hyper-parameters.

4. As shown in Table 5, the contribution of "core-periphery" may be trivial.

5. Given the aforementioned issues, "a large margin of performance improvements" is not a proper reason to accept this paper.

**Questions:**

See weaknesses.

---

### Official Review · Reviewer_mFhM · 2023-10-30

**Soundness:** 2 fair
**Presentation:** 1 poor
**Contribution:** 2 fair
**Rating:** 3
**Confidence:** 4

**Summary:**

The paper presents a ViT framework, RCCT (robust core-periphery constrained transformer), for unsupervised domain adaptation that utilizes a modified self-attention mechanism inspired by the "core-periphery" principle in neuroscience. Specifically, this boils down to two steps: 1) for each patch token in the transformer, calculate the "coreness" or the degree of domain invariance using a patch-level discriminator, 2) mask the standard self-attention with these values to ensure more focus on these domain invariant patches for better adaptation. The framework also uses a "latent feature interaction" mechanism which is a form of mixup in the feature space. A global feature-level discriminator is also added to encourage learning of domain-invariant features via adversarial learning, along with a mutual information maximization loss to achieve better pseudo-labels. Experiments are conducted on four benchmarks: Office31, OfficeHome, DomainNet, and VisDA, with RCCT achieving improvements in the range of 0.8-3% over existing methods.

**Strengths:**

While unsupervised domain adaptation (UDA) is quite a well established problem, the majority of the algorithms has been built on CNNs. With the ever increasing popularity of transformers in all facets of computer vision, it is important to study how these backbones work with UDA, especially with recent studies discovering the robustness of ViTs to OOD samples. Thus, the problem this paper is solving is quite an important and relevant one. The neuroscience inspired methodology is also new in this domain.

**Weaknesses:**

1. The presentation of the paper is very poor, which made it hard to understand the core contributions
- For a paper which involves modifications of an architecture (ViT), it is crucial to introduce the components of the basic architecture to contextualize the modifications and be self-contained. This paper does not do this.
- There is no proper organization of the methodology section. Sec 3.4 uses a graph constructed in Sec 3.3.1, however, the relevance of the graph is never mentioned in the latter. This leads to a "back and forth" narrative structure.
- The equations and notations are all over the place. Eq. 3 uses the indices *h* and *b* for summation, however, the rest of the equation uses some other indices. *C* is mentioned as a scalar, but somehow the matrix transpose operation is used. Similarly, *f_ip* is used to dennote the *r*-th token of *i*-th image - what is *r* here?
- Eq. 9,10 includes the V inside the softmax.

There is no clear structure and mathematical notation is not used or explained properly.

2. In the latent-feature interaction (LFI), the operation defined in Eq. 8 does not make sense to me. It involves mixing the features of two distinct images without mixing the labels of the instances. This disregard of the class information is confusing. The authors draw a weak analogy to noise in brain processing, but no proper grounded explanation is given beyond "The LFI operation helps generate robust CP graphs." How?

3. The ablation studies in Table 5 shows that a huge chunk of the performance improvements come from the mutual information loss, which has been used before in multiple UDA papers [a]. The ablation for the global domain discriminator is also not present.

4. As shown in Fig 3, different UDA setups require *very* different mixture values for the LFI operation. It is not clear how these values are chosen.

5. None of the baselines are discussed in the text or cited. I have no idea what these methods are since they are simply acronyms in the paper.

[a] Do We Really Need to Access the Source Data? Source Hypothesis Transfer for Unsupervised Domain Adaptation, ICML 2021

**Questions:**

1. What are the baselines? How do they differ from your method?
2. Why is there no related works section on UDA?
3. What is the meaning of the LFI operation?
4. Update Fig 1 to give more context in terms of the transfomer, the current one explains nothing.

---

### Official Review · Reviewer_5i8Q · 2023-11-01

**Soundness:** 3 good
**Presentation:** 2 fair
**Contribution:** 3 good
**Rating:** 5
**Confidence:** 4

**Summary:**

This paper focuses on unsupervised domain adaptation (DA) where the goal is to transfer the knowledge from a labeled source domain to an unlabeled target domain. They draw motivation from brain structure and function, specifically from core periphery (CP) principle and noisy brain concept to propose robust core-periphery constrained transformer (RCCT) for DA. In RCCT, the self-attention operation is modified through an adaptively learned weighted graph with the CP structure. Specifically, they divide the image patches into core and periphery patches where core patches are more domain-invariant and periphery patches are more domain-specific. Based on the noisy brain concept, they propose latent feature interaction (LFI) where mixup is done in the latent space to improve the stability and robustness of the CP graphs. Finally, they evaluate the proposed RCCT on standard DA benchmarks.

**Strengths:**

* The proposed methods seem fairly well motivated, interesting, and novel. The idea of suppressing domain-specific information flow and encouraging domain-invariant information flow is intuitive.

* The proposed RCCT achieves consistently good improvements across all benchmarks.

* The paper is well-written and easy to follow.

**Weaknesses:**

* Important ablation studies are missing
    * There are no ablation studies on the different loss terms involved. Hence, it is unclear which losses contribute more or less to the performance. For example, is the global discriminator required if the patch discriminator is already performing a similar task?
    * In Table 5, CP and LFI are added to a baseline that already uses SCM (and SCM gives the most improvement compared to CP and LFI). But we also need to check how well CP and LFI can work on their own since the motivation for these components does not mention the necessity of SCM.
    * Hence, it seems that the self-clustering loss (that is used by many different works) contributes the most while adversarial training with discriminators is less effective (and more complicated to optimize). This potentially weakens the claims of how important the core periphery constraints are.

* Regarding novelty of LFI
    * It seems very similar to mixup at the feature-level (for example in [W1]). In general, mixup is also a fairly common training augmentation. Please compare the similarities and differences since it hurts the technical novelty of LFI.

* Overall framework seems to be complicated
    * Since the method involves adversarial training with a global discriminator and a patch-wise discriminator, the method might be difficult to tune or optimize. Further, there are no hyperparameter sensitivity analyses for the loss weighting hyperparameters like $\alpha, \beta, \gamma$.
    * As per the Supplementary, different hyperparameters are used for different benchmarks, so it is unclear how much tuning is required for a new benchmark.

* Some important details of the approach are unclear
    * Page 5: there is no $r$ in the equation for $C(f\_{ip})$, so it is unclear to me.
    * Eq. 3: $h$ and $b$ are not involved in the inner terms and the dimensions of $C(f\_{ip})$ are unclear. From the original definition, $C(f\_{ip})$ seems to be a scalar but a vector inner product is being computed in Eq. 3.
    * Eq. 4: what is $M(i, j)$? How is it computed? Is it actually Eq. 3 which defines it as $M\_{cp}$?
    * Eq. 4: why is the threshold chosen to be 0.5? How sensitive is the approach to changes in this threshold?
    * Sec. 3.3.2: what is the meaning of “sequentially chosen”?

### References

[W1] Lim et al., “Noisy Feature Mixup”, ICLR22

**Questions:**

* Please see the weaknesses section.

* The high-level motivation is similar to [W2] which tries to separate the processing of domain-invariant and domain-specific factors in a transformer architecture. While it may be considered concurrent work for now, please compare the similarities and differences with this work in later drafts.

* Minor comments
    * Page 3: please add $=$ after $D\_s$ and $D\_t$ in the first paragraph of Sec. 3.1.
    * Below Eq. 2: typo in $p^\text{th}$ patch of …
    * Page 5: typo in “patches that can easily deceive the patch dominator is…” → “patches that can easily deceive the patch discriminator are…”.
    * Page 5: $H$ is used for both number of heads and entropy. Please change one to avoid confusion.
    * Eq. 4: Why not just use \sqrt and ^2? It would be better to have more formal notations. This seems more code-like.
    * Eq. 3 (and in general): use \top instead of T for transpose.
    * Sec. 3.3.2: use \cite instead of \citet.
    * $r$ was used for token index in Sec. 3.3.1 and for layer index in Sec. 3.4. Please use different notations to avoid confusion.
    * Eq. 9 and 10: Use \left and \right with the brackets to improve formatting.

### References

[W2] Sanyal et al., “Domain-Specificity Inducing Transformers for Source-Free Domain Adaptation”, ICCV23